# Peptide Self-Assembly Facilitating DNA Transfection and the Application in Inhibiting Cancer Cells

**DOI:** 10.3390/molecules29050932

**Published:** 2024-02-21

**Authors:** Jingyu Wang, Min Ye, Baokuan Zhu

**Affiliations:** 1School of Biomedical Engineering and Technology, Tianjin Medical University, Tianjin 300070, China; 18504891222@163.com; 2College of Pharmacy, Southern Medical University, Guangzhou 510280, China; yemin@cuhk.edu.cn

**Keywords:** transfection, self-assembled peptide, cancer cells

## Abstract

Non-viral vectors have been developing in gene delivery due to their safety and low immunogenicity. But their transfection effect is usually very low, thus limiting the application. Hence, we designed eight peptides (compounds **1**–**8**). We compared their performances; compound **8** had the best transfection efficacy and biocompatibility. The transfection effect was similar with that of PEI, a most-widely-employed commercial transfection reagent. Atomic force microscope (AFM) images showed that the compound could self-assemble and the self-assembled peptide might encapsulate DNA. Based on these results, we further analyzed the inhibitory result in cancer cells and found that compound **8** could partially fight against Hela cells. Therefore, the compound is promising to pave the way for the development of more effective and less toxic transfection vectors.

## 1. Introduction

Gene delivery plays an important role in many aspects, such as gene therapy, mechanism study and vaccine research [1,2,3,4]. Although gene therapy has suffered many setbacks before [5,6], some related drugs have been approved in the last decade [7,8,9,10]. The authorized drugs can be applied in many diseases, such as monogenetic disorder and cancers [11,12]. As we all know, viral vectors are most commonly used in gene therapy for their high transfection efficiency [13,14,15,16]. Non-viral vectors are another type of vector still being researched because they can overcome the drawbacks that exist in the viral vectors, such as high immunogenicity and risk. A few classes of non-viral vectors have been authorized for extensive use [17,18]. But their low transfection efficiency and high toxicity are still the major reasons to constrain their applications [19]. Therefore, synthetic carriers with high efficiency and good biocompatibility are the research direction in gene therapy. 

Self-assembled peptides are used in a variety of studies because of their high effect and good biocompatibility [20,21,22]. Although the application of the peptides in gene transfection is still in its infancy, some progress has been made. For example, the vectors of peptides can be employed in the diseases of tumor, HIV and cystic fibrosis [23,24,25,26]. Herein, we designed eight peptides as the transfection vectors. One of the vectors had the highest transfection efficacy, equal with that of PEI, a most common commercial transfection agent for experimental applications [24]. We then tested the application of the self-assembled peptide in killing cancer cells. The results showed that it had the ability to fight against Hela cells.

## 2. Results

### 2.1. Subsection

#### 2.1.1. Synthesis of the Peptides

In this paper, we generated Nap-GFFHHHHHHYIGSR-NH_2_ (Nap-GFFH_6_YIGSR-NH_2_, compound **1**, Figure 1). Principles of the design are as follows: (1) Nap-GFF can trigger self-assembly to encapsulate nucleic acid and prevent DNA degradation in lysosomes [27]. (2) H_6_YIGSR has good transfection efficiency. H6 can promote lysosomal escape. YIGSR can bind to laminin in the extracellular matrix, promoting the combination of gene vector and the cells [28]. The capping group, -NH_2_, is to facilitate the binding to DNA, which carries the negative charges. Enzyme-catalyzed assembly is a preferable method to improve biological performance [25,29]. 

Therefore, we designed Nap-GFFHHHHHHY(p)IGSR-NH_2_ (Nap-GFFH_6_Y(p)IGSR-NH_2_, compound **2**, Figure 1). The number of “F” determines the assembly of the compounds [30]. Therefore, more compounds were designed as follows: Nap-GHHHHHHY(p)IGSR-NH_2_ (Nap-GH_6_Y(p)IGSR-NH_2_, compound **3**, Figure 1) and Nap-GFHHHHHHY(p)IGSR-NH_2_ (Nap-GFH_6_Y(p)IGSR-NH_2_, compound **4**, Figure 1). D-amino acid-based peptide is more stable and has better biological performance [31]; hence, Nap-G^d^FHHHHHHY(p)IGSR-NH_2_ (Nap-G^d^FH_6_Y(p)IGSR-NH_2_, compound **5**, Figure 2) was synthesized. The experimental data showed that the number of “H” affected the transfection efficacy [32]. In consequence, Nap-G^d^FHHHHHHHY(p)IGSR-NH_2_ (Nap-G^d^FH_7_Y(p)IGSR-NH_2_, compound **6**, Figure 2) and Nap-G^d^FHHHHHHHHY(p)IGSR-NH_2_ (Nap-G^d^FH_8_Y(p)IGSR-NH_2_, compound **7**, Figure 2) were synthesized. It has been confirmed that fluorinated compounds can promote transfection [33,34]. Therefore, Nap-G^d^F(f)HHHHHHHY(p)IGSR-NH_2_ (Nap-G^d^F(f)H_7_Y(p)IGSR-NH_2_, compound **8**, Figure 2) was designed.

#### 2.1.2. Fluorescence Microscope Results

We then tested the transfection effects of the peptides. The compounds **1**–**8** were dissolved in DMEM at the concentrations of 1.5 mg/mL, 0.75 mg/mL, 0.375 mg/mL, 0.18 mg/mL and 0.09 mg/mL, respectively. After they were mixed with the plasmid and alkaline phosphatase (AP), the mixtures assembled overnight at 4 °C. The profile of the plasmid is shown in Figure 1A. The plasmid carried the gene that expresses green fluorescence protein. Therefore, if the plasmid can enter the cell to express the protein, it can be directly observed under the fluorescence microscope. The transfection effects of the compounds were detected by fluorescence microscope and flow cytometry (Figure 1 and Figure 2). The optimal concentration of the compounds for transfection was different. For example, compounds **1**, **2** and **4**–**7** at the concentration of 0.375 mg/mL had the optimal transfection efficiency, compared with that of compounds **3** and **8** at 0.75 mg/mL and 0.18 mg/mL, respectively, (fluorescence microscope results; did not show in the figures). Figure 1 and Figure 2 showed the optimal transfection effects of the compounds. As shown in Figure 1B, a single plasmid had little ability to enter the cells to express proteins. As shown in Figure 1C,D, the transfection efficiency of compound **2** was better than that of compound 1, proving the effect of enzyme-digested assembly. Appropriate assembly was the key element to acquire good performance of the nanomaterials. One phenylalanine (F) in compound **4** had the optimal transfection efficacy, compared with that of compounds **2** and **3**, as shown in Figure 1D,E and Appendix A. Just as we suspected, D-amino acid improved the transfection result, as shown in Figure 1E,F. As for the number of the histidine (H), the experimental data showed that seven H was the best (Figure 1F,G and Appendix A). As shown in Figure 1G,H, fluorinated compound had good performance in gene transfection. Figure 1C–H shows the gradually increasing transfection efficacy; the corresponding compounds were **1**, **2**, **4**, **5**, **6** and **8**, respectively. So the following experiments mainly focused on these compounds. As PEI is the most-commonly-used commercial transection reagent in vitro, we then compared the transfection result of PEI with that of compound **8**. As shown in Figure 1H,I, there was little difference, demonstrating the good performance of compound **8** in gene transfection.

#### 2.1.3. Flow Cytometry Results and Cytocompatibility

To further verify the above conclusion, we employed flow cytometry. As shown in Figure 2A, there were two peaks: the first one representing the background (no transfection cells) or the cells with very low protein expression; the second one representing the transfection results. The conclusion was consistent with that in Figure 1, that is, compounds **1**, **2**, **4**, **5**, **6** and **8** had a gradually increasing transfection effect and compound **8** had a similar transfection effect with PEI. The corresponding statistical data were also shown in Figure 2B,C; they exhibited median fluorescence intensity (MFI) and the positive enhanced-GFP (eGFP) cells. All the results demonstrated that compound **8** was applicable to be as a gene vector. To assess the transfection performance, cell viability was also evaluated. As shown in Figure 2D, cell viability of compound **8** (>90%) showed no significant difference with the control. According to the standard of United States Pharmacopeia (USP), the cytotoxicity was grade 1, ref. [35], attesting to the good cytocompatibility of compound **8** to be used in gene transfection.

#### 2.1.4. AFM Image and the Inhibitory Experiment

We then researched the reason for the good transfection effect and found that compound **8** and the eGFP plasmid could assemble, as shown in Figure 3A. Based on the good performance of compound **8**, we further tested the effect of this compound against cancer cells. Apoptin is a kind of protein that is well known for the inhibition of cancer cells and innocuous to normal cells. Compared with translocation into the nucleus of cancer cells, the apoptin protein is prone to be retained in cytoplasm in normal cells, causing the obvious different inhibitory effect. Hence, we used the apoptin plasmid for the further experiment. As shown in Figure 3B, no matter at what concentration (1.5 mg/mL, 0.75 mg/mL, 0.375 mg/mL, 0.18 mg/mL and 0.09 mg/mL) of compound **8**, cell viability had no obvious difference with that of the control (untreated cells). Furthermore, the cytotoxicity was zero or one, according to standard of USP, [35] verifying the good biocompatibility of compound **8** to Hela cells. After compound **8** and the apoptin plasmid assembled, the hybrid could kill Hela cells, especially at high concentrations such as 1.5 mg/mL and 0.75 mg/mL. Cell viability at these concentrations was 66.5% and 71.6%, respectively. These results testified the peptide 8 could partially kill cancer cells in gene therapy.

## 3. Discussion

To broaden the application of non-viral vectors, we designed eight peptides for transfection. The experimental data showed that peptide 8 had the best transfection effect, as shown in Figure 4. PEI is a widely used reagent in transfection and the efficiency of compound **8** was comparable with PEI. The high transfection effect might have been because the peptide assembled to encapsulate DNA. High transfection efficiency and good biocompatibility of compound **8** might be applicable in further gene therapy. Therefore, we used it in fighting against Hela cells and it partially inhibited the cells (Figure 4). Based on the previous experiments, compounds **3** and **7** had a relatively low transfection effect. This might be due to the weak assembly ability. So peptide 8 might have better packing ability due to its appropriate assembly and this made the transfection activity good. D-peptide did not degrade easily. Fluorocarbon had the property of lipophobia and hydrophobicity, [34] which might make peptide 8 very stable in its life entity. The compound containing polyhistidine could potentiate the lysosomal escape. These properties might be the reasons for more abundant protein expression and the good anticancer activity of peptide 8. In conclusion, the self-assembled peptide had good transfection performance and it might be a promising nanomaterial in gene therapy in vivo in the future.

## 4. Materials and Methods

### 4.1. Chemicals and Materials

Fmoc-amino acids, Nap, Fmoc-Tyr(H_2_PO_3_)-OH, Fmoc-d-4-fluorophenylalanine and o-benzotriazol-1-yl-*N*,*N*,*N*′,*N*′-tetramehtyluronium hexafluorophosphate (HBTU) were acquired from GL Biochem. (Shanghai, China). Rink resin (0.9 mmol/g) was obtained from Nankai University resin Co., Ltd. (Tianjin, China). *N*,*N*-Diisopropylethylamine (DIEA) was obtained from Energy Chemical (Shanghai, China). Alkaline phosphatase (ALP) was acquired from TaKaRa biotechnology Corporation (Beijing, China). Dulbecco’s modified Eagle’s medium (DMEM) and penicillin/streptomycin were purchased from Gibco Corporation (New York, NY, USA). Fetal bovine serum (FBS) was purchased from BI Corporation (Haemek, Israel). 3-(4,5-Dimethylthiazol-2-yl)-2,5-diphenyltetrazolium bromide (MTT) was purchased from Solarbio (Beijing, China). Commercially available reagents and solvents were used directly, unless noted otherwise. 

### 4.2. General Methods

HPLC was conducted on a LUMTECH HPLC (Moorestown, NJ, USA) instrument using a C_18_ RP column with MeOH (0.05% of TFA) and water (0.05% of TFA) as the eluents. LC–MS was conducted on the LCMS-20AD instrument from Shimadzu (Kyoto, Japan). Gene transfection was performed by the fluorescent microscope (Leica, Wetzlar, Germany). Gene transfection efficiency was performed by flow cytometer (East Rutherford, NJ, USA). AFM images were generated on the Bruker system (Billerica, MA, USA). MTT was measured by the Thermo Scientific Microplate Reader (Waltham, MA, USA). 

### 4.3. Syntheses and Characterizations

#### 4.3.1. Peptide Synthesis

The peptide derivatives were prepared by solid-phase peptide synthesis (SPPS) using rink resin. The corresponding N-Fmoc-protected amino acids with side chains that were properly protected were used. Removing the protective group of the resin exposed the amino group. Then, the C-terminal of the first N-Fmoc-protected amino acid using O-(Benzotriazol-1-yl)-*N*,*N*,*N*′,*N*′-tetramethyluroniumhexafluorophosphate (HBTU) as the coupling reagent could be loaded on the rink resin. DIEA was also used to adjust the pH. Then, 20% piperidine in anhydrous *N*,*N*’-dimethylformamide (DMF) was used during deprotection of the Fmoc group, exposing the amino group. Then, the next Fmoc-protected amino acid was coupled to the free amino group using HBTU as the coupling reagent and DIEA as the base. The growth of the peptide chains was according to the established Fmoc SPPS protocol. After the last coupling of Nap, rink resin was washed three times by DMF, followed by three steps of washing using DCM. The peptide derivatives were cleaved using 95% of trifluoroacetic acid in DCM for 30 min. The cleavage reagent was removed by a rotary evaporator 5 times (adding DCM when the reagent did not reduced). A small amount of ether was added to the oil like liquid. The ether was dried and the yellowish solids were obtained (compounds **1**–**8**). The peptides were then purified by HPLC and the molecular weights were tested by LC–MS.

#### 4.3.2. Gene Transfection

Compounds **1**–**8** were dissolved with DMEM at the concentrations of 1.5 mg/mL, 0.75 mg/mL, 0.375 mg/mL, 0.18 mg/mL and 0.09 mg/mL, respectively. Then, 75 μL of the compound was mixed with 0.2 μg of the plasmid. Compounds **2**–**8** had a further 0.75 U of alkaline phosphatase (10 U/mL) added and were assembled with plasmid DNA overnight in a 4-degree refrigerator. Plasmids of the same mass were added to PEI for 30 min as the positive control, and a single plasmid was served as the negative control. 293T cells were digested with trypsin and laid on 96-well plates with 1 × 10^5^ cells per well. After 12 h, the plasmids with the vectors were added into the cells. After 5 h, the solutions were pipetted out and the complete culture medium (DMEM and FBS) was added. The transfection results were observed 72 h later.

#### 4.3.3. Flow Cytometry

The compounds were dissolved and gene transfection was performed as described above. After 72 h incubation, cells in each well were digested separately with trypsin and then fixed with 4% paraformaldehyde. The cells were then filtered with mesh, ensuring the cell density was 1 × 10^5^–5 × 10^5^/mL. Flow cytometry was then used to detect the gene transfection efficiency.

#### 4.3.4. Cytotoxicity Assay after Transfection

Compounds with plasmid DNA were assembled overnight in a 4-degree refrigerator. Plasmids of the same mass were added to PEI for 30 min. 293T cells were digested with trypsin and laid on 96-well plates with 1 × 10^5^ cells per well. After addition of the compounds with plasmid DNA for 5 h, the mixtures were pipetted out and the complete culture medium was added. Untreated cells were used as the control. After 72 h, the medium was pipetted out and the fresh culture medium was added mixed with 10% of MTT. The cells were incubated for 4 h in a cell incubator, followed by adding DMSO. The OD values were measured by the microplate reader at 490 nm.

#### 4.3.5. AFM 

(atomic force microscope)**:** Compound **8** was mixed with the eGFP plasmid and then alkaline phosphatase (10 U/mL) was added. The mixture was co-cultured overnight in a 4-degree refrigerator. The sample was prepared as a single layer on a mica sheet and could be viewed directly under an AFM. 

#### 4.3.6. Cell Viability after Inhibitory Experiment

Compound **8** at different concentrations (1.5 mg/mL, 0.75 mg/mL, 0.375 mg/mL, 0.18 mg/mL and 0.09 mg/mL) was mixed with the apoptin plasmid and co-cultured overnight in a 4-degree refrigerator. Hela cells were digested with trypsin and laid on 96-well plates with 1 × 10^4^ cells per well. After 12 h incubation, compound **8** with/without the plasmid DNA (apoptin) was added. Untreated cells were used as the control. After 5 h, the mixture was pipetted out and the complete culture medium was added. After 48 h, the medium was pipetted out and the fresh culture medium was added mixed with 10% of MTT. The cells were incubated for 4 h in a cell incubator and then DMSO was added. The OD values were measured by microplate reader at 490 nm.

## Data Availability

Data is unavailable due to privacy.

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
