# Peer review of "Peptide Self-Assembly Facilitating DNA Transfection and the Application in Inhibiting Cancer Cells"

_molecules, 2024, doi:10.3390/molecules29050932_

Round 1

Reviewer 1 Report

Comments and Suggestions for Authors

The manuscript by Jingyu Wang and co-worker present a designed self-assembled peptide (a non-viral vectors) that performed a good transfection performance when complexed with DNA. They compared their proposed peptide nanomaterial performance with the performance of the commercially available PEI polymer.  The author designed, synthetized and characterized various types of peptides and carried on gene transfection, flow cytometry and cytotoxicity assay after transfection experiments.

The manuscript is well craft and hypothesis and experiments well fits in the discussion. Literature is fine. The manuscript is quite interesting, and clearly shows the potential use of moderately short peptides as self-assembled vector for gene delivery.

Basically I recommended its acceptance in molecules in the current form.

Just a small remark:

“The peptide  derivatives were cleaved using 95% of trifluoroacetic acid in DCM for 30 minutes. The  cleavage reagent was removed by a rotary evaporator for 5 times (adding DCM when the  reagent never reduced). A small amount of ether was added to the oil like liquid. Dried 197 the ether and the yellowish solids were obtained (compound 1-8).”

Did the author purified the peptides after cleavage from resin?

Comments on the Quality of English Language

Minor editing of English language required

Author Response

The following are our responses to the comments (in Italics) of the editor and reviewers and the changes (underlined) in the manuscript.

Reviewer #1:

  1. [“The peptide derivatives were cleaved using 95% of trifluoroacetic acid in DCM for 30 minutes. The cleavage reagent was removed by a rotary evaporator for 5 times (adding DCM when the  reagent never reduced). A small amount of ether was added to the oil like liquid. Dried the ether and the yellowish solids were obtained (compound 1-8).”

Did the author purify the peptides after cleavage from resin?]

We thank the reviewer for the suggestion. We purified the peptides after cleavage from resin using HPLC and tested the molecular weights by LC-MS. We have added the details in “Materials and Methods” in the manuscript.

Addition: The peptides were then purified by HPLC and the molecular weights were tested by LC-MS.

Reviewer 2 Report

Comments and Suggestions for Authors

This manuscript reported that the authors designed several peptides, they found on one of them, and 8 shows the best transfection efficacy and biocompatibility. The transfection effect was similar with PEI. AFM showed that it can self-assemble and encapsulate DNA. They further found it could fight against Hela cells.

This is an interesting work for peptide-based DNA transfection, however it can not be accepted at present state, a major reversion should be carried out.

1. The author synthesized several compounds. Why does 8 have good anticancer activity?

2. The enveloping ratio of DNA should be provided.

3. The author provided the killing ability for Hela cells, and the killing ability of other cancer cells should be provided.

4. This paper only provides anti-tumor experiments at the cell level, and anti-tumor experiments at the animal level need to be provided.

Comments on the Quality of English Language

Moderate editing of English language required.

Author Response

The following are our responses to the comments (in Italics) of the editor and reviewers and the changes (underlined) in the manuscript.

Reviewer #2:

  1. [The author synthesized several compounds. Why does 8 have good anticancer activity?]

We are thankful for the comment of the reviewer. Based on the previous experiments, compounds 3 and 7 had relatively low transfection effect. This might be due to the weak assembly ability. So peptide 8 might have better packing ability due to its appropriate assembly and made the good transfection activity. D-peptide didn’t degrade easily. Fluorocarbon had the property of lipophobia and hydrophobicity,[1] which might make peptide 8 very stable in life entity. The compound containing polyhistidine could potentiate the lysosomal escape. These properties might be the reasons for more abundant protein expression and the good anticancer activity of peptide 8. We have added the underlined part above in “Discussion” in the manuscript.

  1. [The enveloping ratio of DNA should be provided.]

We thank the reviewer for the comment. In general, the dosage of the plasmid we used in the manuscript was universal and sufficient to the cells.[2, 3] The nitrogen/phosphate (N/P) ratio (molar ratio of amine groups in peptide 8 to phosphate groups in the plasmid) was 355:1.

  1. [The author provided the killing ability for Hela cells, and the killing ability of other cancer cells should be provided.]

We are thankful for the suggestion of the reviewer. Hela cells are commonly employed in gene transfection and can be the representative in gene therapy.[4-8]

  1. [This paper only provides anti-tumor experiments at the cell level, and anti-tumor experiments at the animal level need to be provided.]

We thank the reviewer for the suggestion. The anti-tumor experiments at the animal level are very important for the research of gene therapy.

[1] H. Wang; Y. Wang; Y. Wang; J. Hu; T. Li; H. Liu; Q. Zhang; Y. Cheng. Self-Assembled Fluorodendrimers Combine the Features of Lipid and Polymeric Vectors in Gene Delivery. Angew Chem Int Ed Engl 2015, 54, 11647-51.

[2] Y. Tian; H. Wang; Y. Liu; L. Mao; W. Chen; Z. Zhu; W. Liu; W. Zheng; Y. Zhao; D. Kong; Z. Yang; W. Zhang; Y. Shao; X. Jiang. A peptide-based nanofibrous hydrogel as a promising DNA nanovector for optimizing the efficacy of HIV vaccine. Nano Lett 2014, 14, 1439-45.

[3] K. Han; Q. Lei; H.-Z. Jia; S.-B. Wang; W.-N. Yin; W.-H. Chen; S.-X. Cheng; X.-Z. Zhang. A Tumor Targeted Chimeric Peptide for Synergistic Endosomal Escape and Therapy by Dual-Stage Light Manipulation. Advanced Functional Materials 2015, 25, 1248-1257.

[4] J. Yang; H.Y. Wang; W.J. Yi; Y.H. Gong; X. Zhou; R.X. Zhuo; X.Z. Zhang. PEGylated peptide based reductive polycations as efficient nonviral gene vectors. Adv Healthc Mater 2013, 2, 481-9.

[5] H.Y. Wang; J.X. Chen; Y.X. Sun; J.Z. Deng; C. Li; X.Z. Zhang; R.X. Zhuo. Construction of cell penetrating peptide vectors with N-terminal stearylated nuclear localization signal for targeted delivery of DNA into the cell nuclei. J Control Release 2011, 155, 26-33.

[6] X.Y. Hu; M. Ehlers; T. Wang; E. Zellermann; S. Mosel; H. Jiang; J.E. Ostwaldt; S.K. Knauer; L. Wang; C. Schmuck. Formation of Twisted beta-Sheet Tapes from a Self-Complementary Peptide Based on Novel Pillararene-GCP Host-Guest Interaction with Gene Transfection Properties. Chemistry 2018, 24, 9754-9759.

[7] Y.C. Ryu; K.A. Kim; B.C. Kim; H.D. Wang; B.H. Hwang. Novel fusion peptide-mediated siRNA delivery using self-assembled nanocomplex. J Nanobiotechnology 2021, 19, 44.

[8] S. Wong; J.A. Kemp; M.S. Shim; Y.J. Kwon. Solvent-driven, self-assembled acid-responsive poly(ketalized serine)/siRNA complexes for RNA interference. Biomater Sci 2020, 8, 6718-6729.

Round 2

Reviewer 2 Report

Comments and Suggestions for Authors

The authors addessed all issues we concerned

Comments on the Quality of English Language

Ok